# Potentially Suitable Geographical Area for *Monochamus alternatus* under Current and Future Climatic Scenarios Based on Optimized MaxEnt Model

**DOI:** 10.3390/insects14020182

**Published:** 2023-02-13

**Authors:** Ruihe Gao, Lei Liu, Lijuan Zhao, Shaopeng Cui

**Affiliations:** 1Department of Forest Conservation, College of Forestry, Shanxi Agricultural University, Jinzhong 030801, China; gaoruihe1989@163.com (R.G.); 13623506306@163.com (L.L.); sxndmy2017@163.com (L.Z.); 2Shanxi Dangerous Forest Pest Inspection and Identification Center, Jinzhong 030801, China

**Keywords:** *Monochamus alternatus*, *Bursaphelenchus xylophilus*, MaxEnt, climate change

## Abstract

**Simple Summary:**

Recently, *Monochamus alternatus* have broken through their original distribution areas, showing a tendency to spread to higher latitudes and successfully colonizing the newly invaded areas. Hence, it is important to analyze the potential global suitable area of *M. alternatus* with the latest occurrence coordinates for the monitoring and scientific prevention and control of *M. alternatus*. The optimized MaxEnt model was first used to predict and analyze the potentially suitable areas for *M. alternatus* on a global scale. The results showed that the main temperature factors influencing the potential distribution of *M. alternatus* are monthly mean temperature difference (Bio2), minimum temperature of the coldest month (Bio6), and mean temperature of the warmest quarter (Bio10) and that the main precipitation factors are annual precipitation (Bio12) and precipitation of the driest month (Bio14). Our results indicated that the current and future potential suitable areas of *M. alternatus* might be distributed worldwide.

**Abstract:**

*M. alternatus* is considered to be an important and effective insect vector for the spread of the important international forest quarantine pest, *Bursaphelenchus xylophilus*. The precise determination of potential suitable areas of *M. alternatus* is essential to monitor, prevent, and control *M. alternatus* worldwide. According to the distribution points and climatic variables, the optimized MaxEnt model and ArcGIS were used to predict the current and future potentially suitable areas of *M. alternatus* worldwide. The optimized MaxEnt model parameters were set as feature combination (FC) = LQHP and β = 1.5, which were determined by the values of AUC_diff_, OR_10_, and ΔAICc. Bio2, Bio6, Bio10, Bio12, and Bio14 were the dominant bioclimatic variables affecting the distribution of *M. alternatus*. Under the current climate conditions, the potentially suitable habitats of *M. alternatus* were distributed across all continents except Antarctica, accounting for 4.17% of the Earth’s total land area. Under future climate scenarios, the potentially suitable habitats of *M. alternatus* increased significantly, spreading to a global scale. The results of this study could provide a theoretical basis for the risk analysis of the global distribution and dispersal of *M. alternatus* as well as the precise monitoring and prevention of this beetle.

## 1. Introduction

The pine sawyer beetle, *Monochamus alternatus* Hope (Coleoptera: Cerambycidae), is considered to be an important and effective insect vector for spreading *Bursaphelenchus xylophilus* (Steiner and Buhrer) Nickle (Nematoda: Aphelenchoididae), one of the most harmful invasive alien species worldwide. In fact, *B. xylophilus* is considered a major pest for forest quarantine internationally [1,2,3]. On average, each *M. alternatus* can carry more than 18,000 *B. xylophilus*, and the maximum number is more than 289,000 [3]. *M. alternatus* is an important trunk-boring pest in forest ecosystems, mainly feeding on *Pinus massoniana*, *P. thunbergii*, *P. densiflora*, and other *Pinus* plants. *M. alternatus* is mainly distributed in tropical, subtropical, and temperate zones in China, such as Taiwan, Fujian, Hunan, Zhejiang, and Sichuan provinces, and the northern boundary is between the Hebei and Shandong provinces. In addition, *M. alternatus* also has a wide distribution in East Asian countries, including South Korea, North Korea, Laos, Vietnam, and Japan (Figure 1) [2,3,4,5,6]. However, in recent years, *M. alternatus* has broken through its original distribution areas, spreading to high latitudes and successfully colonizing newly invaded areas. In 2006, 2018, and 2019, wild populations of *M. alternatus* were found in Shanxi, Liaoning, and Jilin provinces, respectively [7,8,9].

Climate change has affected the natural range, abundance, and dispersal of insect populations worldwide [10,11,12]. Global warming is particularly affecting the growth, development, and distribution of insects, causing insects to expand their distribution areas to higher latitudes and elevations to cope with global warming [13,14,15]. As ectotherms, most insects are very sensitive to changes in the temperature of the external environment [16,17]. Studies indicated that meteorological factors can significantly affect the distribution and development of *M. alternatus*, among which temperature and precipitation are the main climatic factors affecting the distribution of this beetle [18]. Unfortunately, the spread of *M. alternatus* to northern China has laid a solid foundation for the invasion of *B. xylophilus* in northern China, which has seriously threatened the ecological security of pine resources. However, there is relatively little information on the effects of global warming on potentially suitable areas for *M. alternatus*. Hence, to effectively control the global spread and establish early dispersal warning networks worldwide for *M. alternatus* and *B. xylophilus*, it is important to analyze the potential global suitable area of *M. alternatus* with the latest coordinates [19].

Species distribution models (SDMs) can be used to predict the potential distribution areas of specific species and estimate the ecological needs of species by fitting the known natural geographic distribution data of species and the corresponding environmental factors affecting their natural distribution [20,21,22,23,24]. Currently, different SDMs have been developed to predict the potential suitable habitats of specific species based on observed distribution data, including Bioclim, Climex, Climate, Domain, and MaxEnt [25,26,27,28]. Among them, the MaxEnt model is considered an excellent prediction tool due to its high prediction accuracy and has been widely used in the prediction of plants, insects, and other organisms because of its advantages of high simulation accuracy and rapid and easy operation [23,26,29,30,31,32]. In addition, the MaxEnt model can be used to simulate and predict species distribution areas only with the geographical spatial data of the target species [33,34,35,36]. Compared with other SDMs, the advantage of the MaxEnt model is that it has a higher prediction accuracy and reliability when applied to the “existence-only” data of species occurrence. In addition, the MaxEnt model also had good prediction ability when a small amount of species distribution data were available [33,34,35,37,38,39,40,41]. However, previous studies have shown that the default parameters of the MaxEnt model may not be optimal for predicting species distribution [15,40,42,43]. Hence, to avoid overfitting of the MaxEnt model when predicting potential species distribution, which leads to a decrease in species transfer ability, optimizing the parameters of the MaxEnt model is necessary to balance its fitting degree and complexity of the MaxEnt model [40,44].

Previous studies have analyzed and predicted the potential suitable areas for *M. alternatus* in China, suggesting that Liaoning and Jilin are not the main suitable areas for *M. alternatus*; in fact, only when the climate warming temperature continues to increase will these areas become the main suitable areas [45,46]. In Korea, the Climex model predicts that *M. alternatus* will spread from the western and southern coastal areas of the Korean Peninsula to the entire country [47]. The potential global distribution area of *M. alternatus* was much larger than its actual distribution area [48]. However, considering that *M. alternatus* has spread and colonized the northern areas of China, such as Shanxi, Liaoning, and Jilin, in recent years, colder regions of the world seem to be suitable for this beetle under current climate conditions. Therefore, it is necessary to fully understand the latest global potential distribution regions of *M. alternatus* on larger spatial scales.

This study used the optimized MaxEnt model combined with climatic data and the latest global distribution data of *M. alternatus* to predict the current and future distribution of *M. alternatus*. This study aimed to (1) determine the optimal MaxEnt model by displaying the settings of the parameter that affect the performance of the model; (2) evaluate the dominant bioclimatic variables affecting the potential distribution of *M. alternatus*; and (3) predict the potential suitable areas for *M. alternatus* under current and future climatic scenarios on a global scale. Our findings can be used to provide a theoretical basis for risk analysis of the spread of *M. alternatus* worldwide as well as precise monitoring and scientific prevention of this beetle.

## 2. Materials and Methods

### 2.1. Global Occurrence Data of M. alternatus

The worldwide distribution record data of *M. alternatus* were collected from references (https://www.webofscience.com/, accessed on 26 December 2022), public databases of the Global Biodiversity Information Facility (GBIF, https://www.gbif.org/), the Centre for Agriculture and Bioscience International (CABI, https://www.cabi.org/), and the website of the National Forestry and Grassland Administration of China (http://www.forestry.gov.cn/). The obtained distribution point data were then sparsified using the spThin package in R software to ensure that only one distribution point was retained within each 5 km × 5 km range [49]. Finally, 189 distribution record points of *M. alternatus* were obtained for the operation and verification of the MaxEnt model (Figure 1).

### 2.2. Bioclimatic Variables

In this study, 19 historical global bioclimatic variables were downloaded from the Worldclim database (https://www.worldclim.org/) with a spatial resolution of 5 arc-minutes and time range of 1970–2000. Future climate data were obtained from Coupled Model Interaction Project Phase 6 (CMIP6) under the Beijing ClimateCenter Climate System Model 2 Medium Resolution (BCC–CSM2–MR) climate model for SSP126 (sustainability, the most optimistic scenario reflecting RCP2.6 from the fifth report) and SSP585 (fossil-fuel based development or business-as-usual, reflecting RCP8.5), which included the years 2040 (2031−2050 average), 2070 (2061–2080 average), and 2100 (2081−2100 average).

These bioclimatic variables closely affected the growth and development of *M. alternatus*. Using the sampling function of ArcGIS10.7 software, the bioclimatic variables of 189 distribution record points of *M. alternatus* were extracted in this study (Appendix A). To avoid the influence of autocorrelation from multiple linear repeats among the extracted climate variables and avoid the overfitting of MaxEnt [50,51], screening and removal of some of the climate variables were performed to reduce the influence of redundancy on the prediction results. First, we used the extraction analysis tool in ArcGIS software to perform the multicollinearity analysis of each climatic variable; then, the data were analyzed using Pearson correlation in SPSS 22.0 (Appendix A). In order to improve the accuracy of model simulation, the values of climatic variables with correlation values greater than 0.8 could be removed to reduce the influence of overfitting on the model (Figure 2) [52,53]. Finally, eight bioclimatic variables (Table 1) were selected as environmental variables to predict the suitable area for *M. alternatus*.

### 2.3. MaxEnt Model Setting and Selection

The parameters of the MaxEnt model were set as follows: the model output format was selected as “Logistic”, file type as “Asc”, and replication run type as “Subsample”. “Create response curves” and “Do Jackknife to measure variable importance” were selected to analyze the importance of environmental factors. We used the “ENMeval” package in R software to calculate the values of β and FC [15,40,43,54]. This model has five features including linear (L), quadratic (Q), hinge (H), product (P), and threshold (T). To optimize the model, the eight β values were set to 0.5, 1, 1.5, 2, 2.5, 3, 3.5, and 4, and six FC combinations were tested including L, LQ, H, LQH, LQHP, and LQHPT. The package of “ENMeval” in R software was used to verify the above 48 parameter combinations. The Akaike information criterion correction (AICc) was calculated using the Checkerboard 2 method to assess the complexity and fit, and the lowest delta AICc score was applied to the model. The difference between the training and test areas of the receiver operating characteristic curve (AUCdiff) and AICc was used to select the optimal parameter combination for the MaxEnt model [23,42]. Finally, an optimized MaxEnt model was built with the maximum number of iterations set as 5000 to optimize convergence and the maximum number of background points set as 10,000 and repeated 10 times.

### 2.4. MaxEnt Model Evaluation and Analysis

The AUC value was used to test the accuracy of model prediction [15,55,56]. Generally, a larger AUC value indicates better prediction. The AUC values ranged from 0 to 1, and the classification used to assess the accuracy of the MaxEnt results was failing (0–0.6), poor (0.6–0.7), fair (0.7–0.8), good (0.8–0.9), and excellent (0.9–1.0).

The prediction results of the MaxEnt model were imported into ArcGIS software. Moreover, ArcGIS 10.7 software was used to visually transform and reclassify the predicted results, and the Jenks’ natural break method was used to divide the global fitness of *M. alternatus* into four classes: nonsuitability zone (0–0.05), low-suitability areas (0.05–0.22), moderately suitable areas (0.22–0.48), and highly suitable areas (0.48–1.00). Finally, the proportions of different suitable areas were determined and the areas of different suitable areas were calculated.

## 3. Results

### 3.1. MaxEnt Model Optimization and Accuracy Evaluation

In this study, the performance of the optimized MaxEnt model is excellent. The default setting parameters were β = 1, FC = LQHP, AUC_diff_ = 0.020, and ΔAICc = 3.539. After the optimization, the model parameters in this study were set as FC = LQHP, β = 1.5, AUC_diff_ = 0.018, and ΔAICc = 0 (Table 2). The ΔAICc values and AUC_avg_ for 48 different combinations of MaxEnt parameter settings were established using ENMeval (Appendix A). The MaxEnt model showed high prediction performance, with an AUC value of 0.987 ± 0.0002 after model training and 0.987 ± 0.0014 after model testing (Appendix A). Meanwhile, the values of AUC_DIFF_, OR_MTP_, and OR_10_ under the optimized model decreased. Furthermore, the optimal MaxEnt model was selected to repeat 10 times under the current climate conditions. Therefore, this optimized MaxEnt model can be used for the prediction and analysis of the potential suitable distribution of *M. alternatus*.

### 3.2. Relationships between the Distribution of M. alternatus and Bioclimatic Variables

Among the eight bioclimatic variables, Bio14, Bio12, and Bio10 were the top three variables used in the MaxEnt model prediction, with a cumulative contribution rate of 88.9% (Table 1). The results of the jackknife test showed that Bio2, Bio6, Bio14, Bio12, and Bio10 played a huge role in controlling the *M. alternatus* distribution (Figure 2). Therefore, the dominant bioclimatic variables of temperature factors affecting the potential distribution of *M. alternatus* are Bio2, Bio6, and Bio10, and the main precipitation factors are Bio12 and Bio14.

### 3.3. Potentially Suitable Distribution Areas of M. alternatus under Current Climate

The current potentially suitable distribution areas of *M. alternatus* were established according to four grade standards: high suitability, moderate suitability, low suitability, and poor suitability (Figure 3). The suitable habitat area of *M. alternatus* in the world is about 7.52 × 10^6^ km^2^, which accounts for 3.42% of the world’s land area, and the suitable areas of *M. alternatus* are distributed across all continents except Antarctica. Moreover, the areas of highly suitable, moderately suitable, and low-suitability habitats are 1.95 × 10^6^ km^2^, 1.31 × 10^6^ km^2^, and 4.26 × 10^6^ km^2^, accounting for 25.93%, 17.42%, and 56.65% of the total suitable area, respectively. Globally, the highly and moderately suitable habitats of *M. alternatus* are mainly distributed in Asia, including China, North Korea, South Korea, Japan, and the northern part of Vietnam, and are scattered in parts of India, Pakistan, Bangladesh, Myanmar, Nepal, and Bhutan. In addition, there are a small number of moderately suitable habitats for *M. alternatus* along the southeast coast of the United States.

### 3.4. Change in Potentially Suitable Distribution Areas of M. alternatus under the Future Climate

The potentially suitable distribution areas of *M. alternatus* in 2040, 2070, and 2100 were predicted under future climate scenarios of SSP126 and SSP585 (Figure 4). Compared to the current climate scenario, there is a significant increase in the potential global suitable area of *M. alternatus* in future climate scenarios. The new suitable habitats for *M. alternatus* were mainly distributed in the central part of the American continent and Europe; in India, Pakistan, Bangladesh, Vietnam, Myanmar, and Laos in Asia; and Australia in Oceania (Figure 5).

In the 2040s, the potentially highly suitable areas of *M. alternatus* would remain basically unchanged under the SSP585 scenario; however, these areas would increase by 5.64% under the SSP126 scenario. Under the SSP126 and SSP585 scenarios, the medium suitability area will increase by 4.58% and 9.92%, respectively, and the low-suitability area is expected to increase by 39.44% and 15.49%, respectively (Table 3). In the 2070s, the high- and low-suitability areas of *M. alternatus* showed an increase under future scenarios. The moderately suitable area is expected to increase by 6.11% but would decrease by 5.34% under the SSP585 scenario. In the 2100s, the suitable area for *M. alternatus* showed a trend of continuous increase compared to that of the present. The increased suitable area is mainly distributed in northeast China, eastern India, northern Vietnam, and northern Laos in Asia; southern Poland, western Ukraine, and a small part of western Russia in Europe; the central and northern United States in North America; northeastern Australia in Oceania; and the east coast of South Africa and Madagascar in Africa.

## 4. Discussion

In recent years, *M. alternatus* has broken through its original distribution areas, spreading to high latitudes and successfully colonizing newly invaded areas [7,8,9]. Previous studies have shown that climatic variables are important for determining distribution patterns of *M. alternatus* worldwide [10,11,12,18]. However, relatively little information is currently available on the effects of global warming on potentially suitable areas for *M. alternatus*. Hence, it is important to analyze the latest global potential distribution regions of *M. alternatus* on larger spatial scales.

The performance of the optimized MaxEnt model was excellent in this research, which indicates that the prediction results have high reliability [53]. The MaxEnt model established based on the default parameter conditions is not suitable and other setting parameters should be selected for this model [15,40,42,43]. After the optimization, the model parameters were set as FC = LQHP, β = 1.5, AUC_diff_ = 0.018, and ΔAICc = 0, which indicates that the optimized model has high reliability. Moreover, the AUC test showed that the accuracy of the model was excellent and could be used for the prediction and analysis of the potentially suitable distribution for *M. alternatus*. Additionally, the values of AUC_DIFF_, OR_MTP_, and OR_10_ under the optimized model decreased, indicating that the accuracy and stability of the MaxEnt model were improved compared with those before optimization. Compared with previous studies on the prediction of suitable areas for *M. alternatus*, the sampling sites in this study are more comprehensive, including the new invasion sites of *M. alternatus* in high latitudes in recent years such as the Jilin, Shanxi, and Liaoning provinces in northern China.

Clarifying the relationship between insects and environmental variables is important in understanding the ecological needs and spatial distribution of insect species [54]. Previous studies have shown that temperature and precipitation are the dominant climatic factors affecting the growth and spatial distribution of *M. alternatus* [18,19,55,56]. As ectotherms, most insects are particularly sensitive to changes in environmental temperature, and slight temperature changes can further affect the distribution of insect species [10,16,17,31]. In our study, the suitable temperature for the survival and reproduction of *M. alternatus* was 12–25 °C. Moreover, low temperatures in winter are key factors affecting the distribution of *M. alternatus* [3,57]. In this study, the optimum minimum temperature in the coldest month (Bio6) was −25 °C to 23.15 °C, which played an important role in regulating the survival of overwintering larvae. The mean temperature of the warmest quarter (Bio10) had a positive impact on the distribution of *M. alternatus*; however, the optimum value for the monthly mean temperature difference (Bio2) had a negative impact on the *M. alternatus* distribution. Furthermore, precipitation can indirectly affect the distribution of *M. alternatus*, mainly by changing air humidity, soil humidity, and the growth of host plants [53,58]. Studies show that excessive precipitation may significantly reduce the flight ability of *M. alternatus*, thereby affecting its dispersal and diffusion [18,57,59]. The most suitable precipitation range of the annual precipitation affecting the distribution of *M. alternatus* was 115.82–4404.3 mm and that of the precipitation of the driest month was 1.18–191.40 mm.

The results of the MaxEnt model showed that the current and future potentially suitable habitats of *M. alternatus* would be widely distributed globally. The optimized model indicated that the current global potential areas of *M. alternatus* include its current actual distribution area, indicating that this model has high accuracy and reliability [23]. Moreover, the current potential suitable areas are broader than the actual distribution areas of *M. alternatus*. For example, countries such as Vietnam, India, Pakistan, Bangladesh, Myanmar, Nepal, Bhutan, and the United States currently have no distribution areas for wild populations of *M. alternatus*.

Climate change is likely to significantly affect the distribution of *M. alternatus*. The global average temperature will increase by 1.1–6.4 °C at the end of the 21st century [18,60]. As ectotherms, insects are sensitive to changes in environmental temperature [16,17]. The results of this study indicate that the global potential suitable distribution area of *M. alternatus* will increase significantly in the future and that the new suitable habitats will mainly be concentrated in the central American region and in Europe, India, Pakistan, Bangladesh, Vietnam, Myanmar, Laos, and Australia. These results suggest that global warming leads to significant changes in the range of potential habitat areas for *M. alternatus*. Furthermore, in the context of global warming, *M. alternatus* can continue to spread to higher latitudes, higher elevations, and regions with lower winter temperatures [13,14,15]. Therefore, local forestry departments should focus on the distribution areas and effective prevention and control of major diseases and pests represented by *M. alternatus* to avoid more serious losses to forest resources and forest ecosystems.

Furthermore, in this study, the potential suitable distribution area predicted by the MaxEnt model can only represent areas with similar environmental conditions as the current distribution area and may still be different from the actual distribution area of *M. alternatus*. In addition, only 19 bioclimatic variables were selected for the MaxEnt model prediction, and biotic factors affecting the growth and distribution of *M. alternatus*, such as the population of human activities and natural enemies [61,62], were not selected. Therefore, it is necessary to consider both biological and abiotic variables affecting the growth and distribution of *M. alternatus* when predicting suitable areas in the future to obtain more accurate prediction results.

## 5. Conclusions

The optimized MaxEnt model was first used to predict and analyze the potentially suitable areas of *M. alternatus* on a global scale. The optimized model parameters were set as FC = LQHP and β = 1.5, which were determined by the values of AUC_diff_, OR_10_, and ΔAICc. The prediction results indicate that the main temperature factors are the monthly mean temperature difference, the minimum temperature of the coldest month, and the mean temperature of the warmest quarter and that the main precipitation factors are annual precipitation and the precipitation of the driest month. The current and future potential suitable habitats of *M. alternatus* will be widely distributed globally. Under the current climate conditions, the potential suitable areas of *M. alternatus* were distributed in all continents except Antarctica, accounting for 4.17% of the total land area of the world. The potential future suitable areas of *M. alternatus* increased significantly, showing a trend of further spreading to the global scale.

## Figures and Tables

**Figure 1 insects-14-00182-f001:**
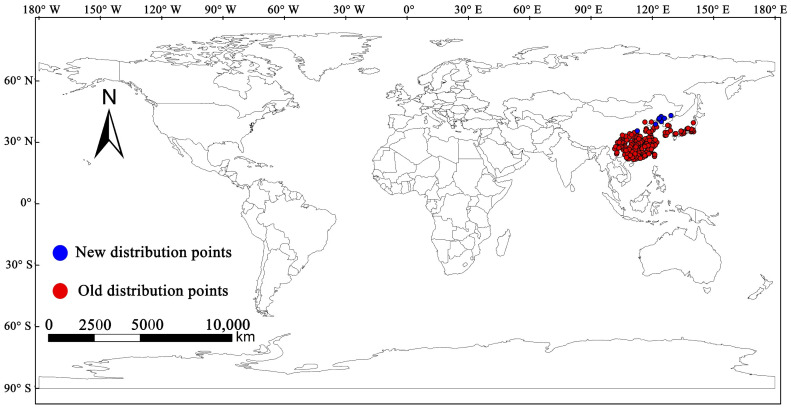
Geographical distribution sampling point of *M. alternatus* at global scale. Red dots represent the traditional distribution points and blue dots represent the new distribution points.

**Figure 2 insects-14-00182-f002:**
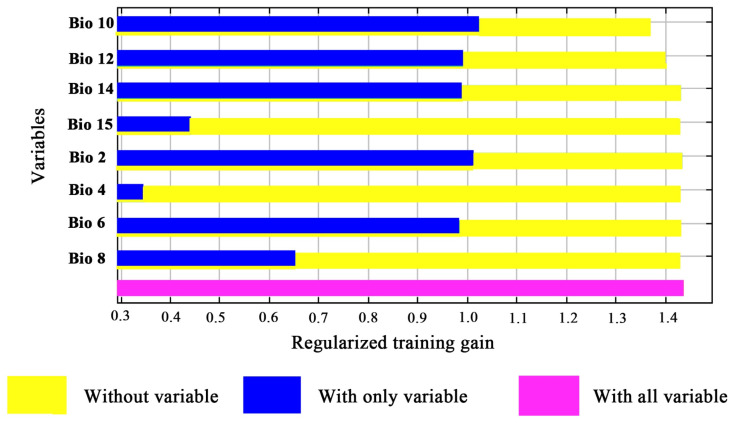
The Jackknife test results on the importance of bioclimate variables.

**Figure 3 insects-14-00182-f003:**
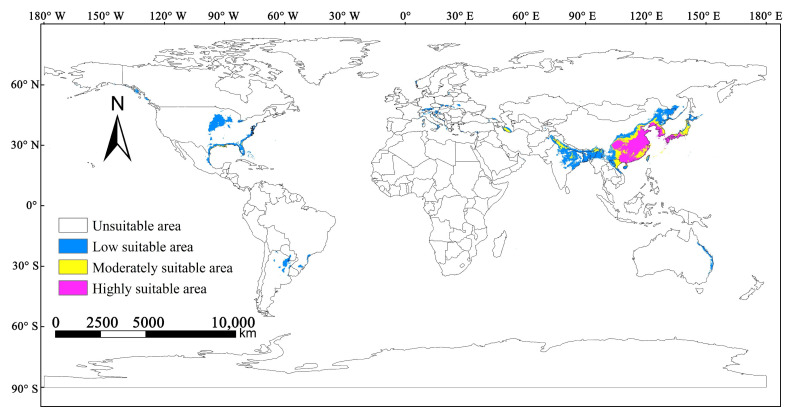
Potential suitable distribution areas of *M. alternatus* at global scale under current climate conditions.

**Figure 4 insects-14-00182-f004:**
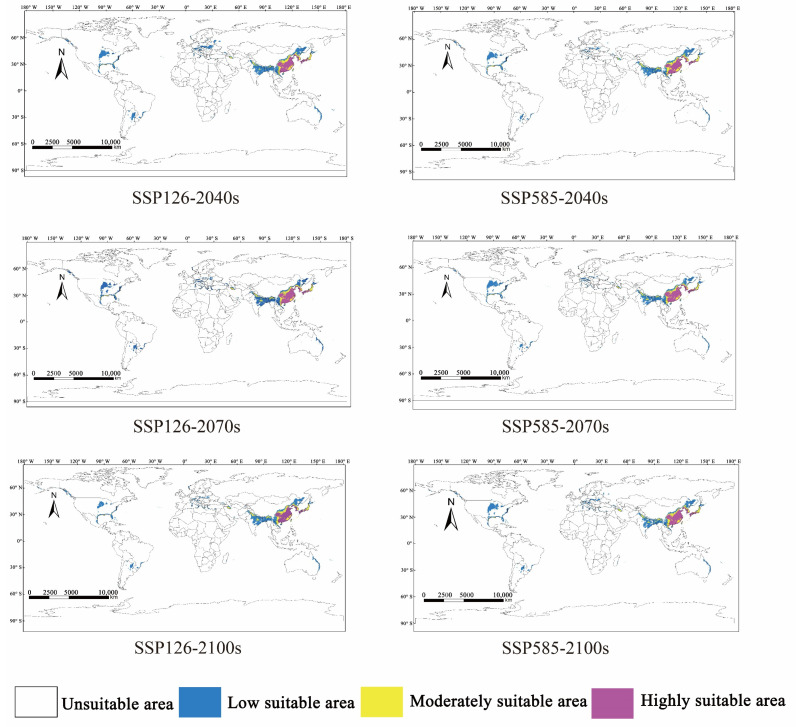
Predicted potential suitable distribution areas of *M. alternatus* at global scale under the future climate conditions of SSP126 and SSP585.

**Figure 5 insects-14-00182-f005:**
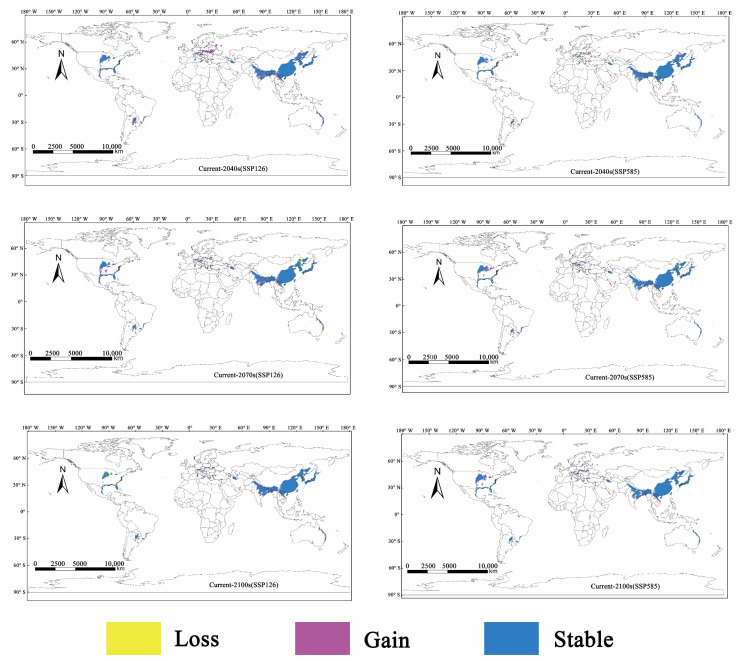
The changes in potential suitable areas of *M. alternatus* in different periods compared with the present.

**Table 1 insects-14-00182-t001:** The contribution rates of selected bioclimatic variables affecting the distribution of *M. alternatus*.

Code	Bioclimatic Variables	Contribution Rate/%
Bio14	Precipitation of driest month	44.5
Bio12	Annual precipitation	24.8
Bio10	Mean temperature of warmest quarter	19.6
Bio6	Min. temperature of the coldest month	3.2
Bio8	Mean temperature of the wettest quarter	3.1
Bio4	Temperature seasonality	3
Bio2	Monthly mean temperature difference	1.3
Bio15	Precipitation seasonality	0.6

**Table 2 insects-14-00182-t002:** Difference in performance of MaxEnt model under default and optimized settings.

	Default	Optimization
β	1.0	1.5
FC	LQHP	LQHP
Mean AUC	0.922	0.923
AUC_DIFF_	0.020	0.018
OR_MTP_	0.042	0.007
OR_10_	0.153	0.146
ΔAICc	3.539	0

**Table 3 insects-14-00182-t003:** The difference in potential suitable areas for *M. alternatus* under current and future climate scenarios.

Decade	Scenarios	Predicted Area (10^6^ km^2^)			Comparison with Current Distribution (%)		
		High	Moderate	Low	High	Moderate	Low
Current	-	1.95	1.31	4.26	-	-	-
2040s	ssp-126	2.06	1.37	5.94	5.64	4.58	39.44
	ssp-585	1.95	1.44	4.92	0	9.92	15.49
2070s	ssp-126	2.13	1.39	5.32	9.23	6.11	24.88
	ssp-585	2.10	1.24	4.97	7.69	−5.34	16.67
2100s	ssp-126	2.01	1.43	4.96	3.08	9.16	16.43
	ssp-585	2.08	1.33	5.31	6.67	1.53	24.65

## Data Availability

Data in this study are available from the corresponding author.

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
