# Peer review of "Potentially Suitable Geographical Area for Monochamus alternatus under Current and Future Climatic Scenarios Based on Optimized MaxEnt Model"

_insects, 2023, doi:10.3390/insects14020182_

Round 1

Reviewer 1 Report

The manuscript entitled "Potentially Suitable Geographical Area for Monochamous alternatus under Current and Future Climatic Scenarios Based on Optimized MaxEnt model” used a distribution modeling approach together with relevant bioclimatic variables (both current and future scenarios) to predict current and potential future suitable areas of M. alternatus on a global scale. Although similar methodologies are common, results of the study and its spatial scale could have useful implications for management actions. However, the work requires major changes before its ready for publication:

Comments and suggestions:

Abstract:

- Line 27:" FC = LQHP”; what does this sand for? "RM", is it regularization multiplier? if so should be written as "β ".

Introduction:

- Introduction doesn't provide sufficient review of the target species in the context of this study particularly at regional or global scale. I suggest adding a new section highlighting the regional/ global Natural distribution of the species.

2. Materials and Methods

-"2.1. Global occurrence records of M. alternatus"; it’s not entirely accurate to use "Global occurrence records"? GBIF records are occurrences and sometimes specimen reports? According to Fig.1 the target species based on the GBIF data only occurs in the East of the globe. Based on the both the available datasets the study should have focused on the Eastern part of the globe? Unless its justified by literature review that the species have been reported in other parts of the world.

- Line 161:  "regularization multiplier (RM) "this should be expressed as" β"

- Line172: "5000" or "500"?

- Line173: "10000" since the model was optimized, was more than 10000 background points were tested? For example ,15000 or any other number?

- Line 176: "AUC" Sometimes AUC alone is not sufficient to evaluate the model performance. Why TSS was not considered alongside the AUC?

3. Results

- Fig.7; Please move this to the Appendix.

4. Discussion

- Line 300-301: Reference is required please.

- Line 307-311: These sentences are already mentioned in the earlier section. Please remove them they are redundant.

- Line 319: “….is not reliable…." please revisit this sentence, default setting, sometimes, doesn't produce accurate models depending on various factors.

- How climatic variables influence the spatial distribution of the target species both for current and the future is not sufficient from the discussion. More in depth description of the influences of the climate change related factors to the dispersal of the species is needed. Furthermore, the discussion should also highlight the benefit and limitations of the applied modeling techniques particularly when it comes to the implications of the current techniques in establishing early warning syste.

Best wishes,

Author Response

Thanks so much for reviewer 1 for his/her positive comments and clear advices, which help improve the quality of the manuscript. We have studied the comments carefully and have revised and corrected the manuscript as reviewer mentioned. Revised portions are marked in yellow green in the revised manuscript. The point to point responses are as following:

Point 1 (L27) : “FC = LQHP”; what does this sand for? "RM", is it regularization multiplier? if so should be written as "β ".

Response 1: Thanks very much for reviewer’s clear advice. In this study, "FC" in L27 is replaced by "feature combination (FC)", which means feature combination. The feature combination selected in this study is “LQHP”. Moreover, the word “regularization multiplier” has been revised to “β” in L28, L155, L157, L182, L259, L322, and Table 2.

Point 2: Introduction doesn't provide sufficient review of the target species in the context of this study particularly at regional or global scale. I suggest adding a new section highlighting the regional/ global Natural distribution of the species.

Response 2: Thanks very much for reviewer’s useful advices. Considering reviewer’s suggestion, the regional and global natural distribution of M. alternatus has been added to the first paragraph in L46-54. Details are as follows:

  1. alternatus is mainly distributed in tropical, subtropical, and temperate zones in China, such as Taiwan, Fujian, Hunan, Zhejiang, and Sichuan provinces, and the northern boundary is between the Hebei and Shandong provinces. In addition, M. alternatus also has a wide distribution in East Asian countries, including South Korea, North Korea, Laos, Vietnam and Japan (Figure 1). However, in recent years, M. alternatus has broken through its original distribution areas, spreading to high latitudes and successfully colonizing newly invaded areas. In 2006, 2018, and 2019, wild populations of M. alternatus were found in Jilin, Shanxi, and Liaoning provinces in northern China, respectively.

Point 3: Global occurrence records of M. alternatus"; it’s not entirely accurate to use "Global occurrence records"? GBIF records are occurrences and sometimes specimen reports? According to Fig.1 the target species based on the GBIF data only occurs in the East of the globe. Based on the both the available datasets the study should have focused on the Eastern part of the globe? Unless its justified by literature review that the species have been reported in other parts of the world.

Response 3: Thanks very much for reviewer’s useful advices. Indeed, it is difficult to accurately determine the occurrence records of M. alternatus on a global scale. That’s mainly because it is difficult for us to field survey whether there is an occurrence of M. alternatus on a global scale. Therefore, we can only use the known distribution data and combine the occurrence records in the database, such as references, public databases of Global Biodiversity Information Facility (GBIF), the Centre for Agriculture and Bioscience International (CABI), and the official website published by the government, to find the known global distribution data of M. alternatus as much as possible to ensure the accuracy of the prediction. In this study, 189 distribution record points of M. alternatus were obtained for the operation and verification of the MaxEnt model.

Point 4 (L172) : "5000" or "500"?

Response 4: In this study, the valus of the maximum number of iterations was 5000, so that the convergence of the MaxEnt model can be better optimized. The relevant references are as follows:

  1. Swets, J.A. Measuring the accuracy of diagnostic systems. Science 1988, 240, 1285-1293.
  2. Xue, Y.; Lin, C.; Wang, Y .; Zhang, Y .; Ji, L. Ecological niche complexity of invasive and native cryptic species of the Bemisia tabaci species complex in China. J. Pest Sci. 2022, 95, 1245–1259.

Point 5 (L173) : "10000" since the model was optimized, was more than 10000 background points were tested? For example ,15000 or any other number?

Response 5: Thanks very much for reviewer’s clear advices. As in most of the literature when using the MaxEnt model for species prediction, the background points were set to 10000 for M. alternatus prediction in this study. We did not select other numbers of background points to optimize the MaxEnt model.

Point 6 (L176) : "AUC" Sometimes AUC alone is not sufficient to evaluate the model performance. Why TSS was not considered alongside the AUC?

Response 6: In this study, we did not only use the value of AUC to evaluate the performance of the MaxEnt model. In fact, the performance of MaxEnt model is determined by a series of parameters, as detailed in Table 2 in MS. Moreover, we are very sorry that I did not understand the meaning of "TSS" proposed by the reviewer. In order to further improve the quality of MS, it is necessary to ask the reviewer to be more specific about this problem. Thank you.

Table 2. The performance of MaxEnt model under default and optimized settings.

Default

Optimization

RM

1.0

1.5

FC

LQHP

LQHP

Mean AUC

0.922

0.923

AUCDIFF

0.020

0.018

ORMTP

0.042

0.007

OR10

0.153

0.146

ΔAICc

3.539

0

Point 7 (Figure 7): For Fig.7, Please move this to the Appendix.

Response 7: Considering reviewer’s suggestion, Figure 7 (Pearson correlation analysis of insect community along the altitudinal gradient in the Guandi Mountain) has been moved to the supplementary materials of the MS. Meanwhile, the relevant explanation and description of Figure 7 in L221-236 has also been deleted from MS. The details of the deleted content are as follows:

Based on the response curve of bioclimatic variables, the probability of the presence of M. alternatus worldwide could be assessed using the MaxEnt model (Figure 7). The optimum value for Bio2 had a negative impact on the distribution of M. alternatus, and the response probability decreased with increasing temperature (Figure 7a). The suitable range of the minimum temperature of the coldest month (Bio6) was -25 °C to 23.15 °C, in which the response probability of M. alternatus increased with the increase in temperature from -25 °C to 10.19 °C, and decreased with the increase in temperature from 10.19 °C to 23.15 °C (Figure 7b). Mean temperature of warmest quarte (Bio10) had a positive impact on the distribution of M. alternatus, and the response probability increased with an increase in temperature from -7.46 to 36.41 °C (Figure 7c). The suitable range of the annual precipitation (Bio12) was 115.82–4404.3 mm, involving a positive relationship with the distribution of M. alternatus for precipitation below 1136.42 mm but a negative response for precipitation greater than 1136.42 mm (Figure 7d). Furthermore, the suitable range of the precipitation of driest month (Bio14) was 1.18–191.40 mm, and the response probability of M. alternatus increased for Bio14 less than 105.17 mm, whereas it decreased at greater than 105.17 mm (Figure 7e).

Point 8 (L300-301) : Reference is required please.

Response 8: Many thanks to the reviewers for their good suggestions. According to the comments of the reviewers, the relevant references [7-9] were placed in the original sentences in L54 and L249. The details are as follows:

  1. Wang, Z.M.; Pi, Z.Q. Hou, B. Monochamus alternatus were found in Jilin Province. Forest Pest and Disease. 2006, 03, 35. (in chinese)
  2. Li, S.W,; LYU, X. L.; Tian, Y.M. et al. Population dynamics of Monochamus alternatus in a typical occurrence area of Pine Wilt Disease in Dalian City. Liaoning Forestry Science and Technology. 2019, 06, 20-22. (in chinese)
  3. Li, Y.X.; Zhang, X. Y. Analysis of invasion expansion trend of Bursaphelenchus xylophilus. Forest Pest and Disease. 2018, 37(05), 1-4. (in chinese)

Point 9 (L307-311) : These sentences are already mentioned in the earlier section. Please remove them they are redundant.

Response 9: Considering reviewer’s suggestion, the details of the deleted content are as follows:

Based on the maximum entropy theory, the MaxEnt model simulates and analyzes the distribution of species when the entropy reaches a maximum under restricted conditions, according to the distribution of the known species and the corresponding environmental variables. This model has the advantages of high simulation accuracy, short operation time, and easy operation and has been widely used in the prediction of potentially suitable areas for plants, insects, and other organisms.

Point 10 (L319) : “….is not reliable…." please revisit this sentence, default setting, sometimes, doesn't produce accurate models depending on various factors.

Response 10: Many thanks to the reviewers for their good suggestions. According to the comments of the reviewers, the word “reliable” has been reivisied to “suitable” in the L257.

Point 11:How climatic variables influence the spatial distribution of the target species both for current and the future is not sufficient from the discussion. More in depth description of the influences of the climate change related factors to the dispersal of the species is needed. Furthermore, the discussion should also highlight the benefit and limitations of the applied modeling techniques particularly when it comes to the implications of the current techniques in establishing early warning system.

Response 11: Thanks very much for reviewer’s clear advices. Considering reviewer’s suggestion, the impact of climate variables on future and current spatial distribution of M. alternatus has been added to L299-306. In addition, the benefits (L82-87) and limitations (L87-92 and L310-315) of the MaxEnt model have been added to MS.

Response 12: Due to the addition and reduction of some references, the order of references in MS has been adjusted to some extent, and we have also marked them in MS accordingly.

Response 13: By reviewing the relevant literature, we know that people with red-green color blindness are able to see blue, yellow, and purple. Therefore, we have added the above three colors in Figures 6,8,9, and 10 (and now Figure 2, 3, 4, 5).

Response 14: Due to the reduction of Figures, the order of Figures in MS has been adjusted to some extent, and we have also marked them in MS accordingly.

Reviewer 2 Report

M. alternatus is the most important vector of PWN. The authors parameterised a MaxEnt model to determine the current and future suitable areas for the beetle, based on current distribution records and climatic data.

This is clearly an important pest, and its potential future distribution has strong implications for the risk PWN poses globally. I have some concerns about the quality of the distribution data used for the model – if the data aren’t sound the model will be similarly unreliable. The authors should comment on the quality of the distribution data, particularly at the edges of the beetle’s current range. The authors also refer to enormous changes in suitable distribution which don’t seem to be reflected in the figures. The authors should clarify how the MaxEnt model predictions compare to the previous CLIMEX model and clarfiy why this model is a significant improvement over the existing model. Finally, I would find it quite interesting – and would feel more confident that the model was working - if the authors were able to re-run the model in order to compare the current distribution with the distribution of a decade? 2 decades? ago, to see whether the model was able to predict the recent changes in distribution / spread to higher latitudes.

Specific points:

L17-19 the “Bio” variables are meaningless unless you read the table – rephrase to refer to the actual variables

L29 “quarter”

L42 rephrase – hyperbole

L62-65 and Figure 1 – I would prefer this to be a detail of the current distribution with the new sites in a different colour or shape

L116-126 how reliable are these data? How do you know the beetle isn’t present outside this area and just hasn’t been picked up?

L147 you need to say what all of these variables are or eliminate the figure

L151 I would prefer to see all of the tested bioclimatic variables including the ones that were not significant.

I would put figures 3, 4, and 5 in supplementary info with more information about what the abbreviations mean – figures should be standalone

L215 “quarter” – repeated throughout text, please check all

L223-236 this is just describing the figure – should be cut down

Figure 8 overlay current distribution points? And have a detail showing current distribution?

L335-341 repetition of results – rephrase / cut down and include direction?

L345 give direction of results – this is an issue throughout

L346-348 clarify

L367-369 Is the figure wrong? I find it very confusing that the figures for current and future distributions look essentially the same

L394-396 biotic factors?

Author Response

We are greatly appreciating for reviewer 2for his/her clear advices and dedicated efforts, which help improve the quality of the manuscript. Those comments are all valuable and very helpful for revising and improving our manuscript, as well as the important guiding significance to our researches. We have studied the comments carefully and have revised and corrected the manuscript which we hope meet his/her approval. Revised portions are marked in light blue in the revised manuscript. The point to point responses are as following:

Point 1 (L17-L19) : the “Bio” variables are meaningless unless you read the table – rephrase to refer to the actual variables.

Response 1: Thanks very much for reviewer’s clear advice. According to the comments of the reviewers, the full names of the variables Bio2 (monthly mean temperature difference), Bio6 (minimum temperature of the coldest month), Bio10 (mean temperature of warmest quarter), Bio12 (annual precipitation), and Bio14 (precipitation of driest month) have been added to L17-20 of MS.

Point 2 (L29) : “quarter”.

Response 2: We are sorry that this is a mistake in the spelling of the word, thanks very much for reviewer’s useful advices. The word “quarte” has been revised to “quarter. In addition, we have gone through the manuscript again to check the misspelling of "quarter" and corresponding modifications have been made in L18, L280, L325, and Table1.

Point 3 (L42) : rephrase – hyperbole

Response 3: Many thanks to the reviewers for their useful suggestions, which are very helpful to improve the quality of MS. The word “one of” has been added in the original sentence of L41, so that it can avoid exaggerating the meaning of the sentence.

Point 4 (L62-65 and Figure1) : I would prefer this to be a detail of the current distribution with the new sites in a different color or shape

Response 4: Thanks very much for reviewer’s useful advices. Considering reviewer’s suggestion, the new occurrence geographical distribution sampling point of M. alternatus has been marked with blue color in Figure 1, and the corresponded explanation has been added in the caption. Moreover, the resolution of Figure 1 has been improved in the MS.

Point 5 (L116-L126) : how reliable are these data? How do you know the beetle isn’t present outside this area and just hasn’t been picked up?

Response 5: Thanks very much for reviewer’s useful advices. Indeed, it is difficult to accurately determine the occurrence records of M. alternatus on a global scale. That’s mainly because it is difficult for us to field survey whether there is an occurrence of M. alternatus on a global scale. Therefore, we can only use the known distribution data and combine the occurrence records in the database, such as references, public databases of Global Biodiversity Information Facility (GBIF), the Centre for Agriculture and Bioscience International (CABI), and the official website published by the government, to find the known global distribution data of M. alternatus as much as possible to ensure the accuracy of the prediction. In this study, 189 distribution record points of M. alternatus were obtained for the operation and verification of the MaxEnt model.

Point 6 (L147) : you need to say what all of these variables are or eliminate the figure

Response 6: Thanks very much to the reviewer for your clear suggestions. According to reviewer’s advice, the full names of 19 bilclimatic variables have been added into the captions of Figure 2. Details are as follows:

Bio1 represents the Annual mean temperature, Bio2 represents the Monthly mean temperature difference, Bio3 represents the Isothermality, Bio4 represents the Temperature seasonality, Bio5 represents the Max temperature of warmest month, Bio6 represents the Min. temperature of the coldest month, Bio7 represents the Temperature annual range, Bio8 represents the Mean temperature of the wettest quarter, Bio9 represents the Mean temperature of driest quarter, Bio10 represents the Mean temperature of warmest quarter, Bio11 represents the Mean temperature of coldest quarter, Bio12 represents the Annual precipitation, Bio13 represents the Precipitation of wettest month, Bio14 represents the Precipitation of driest month, Bio15 represents the Precipitation seasonality, Bio16 represents the Precipitation of wettest quarter, Bio17 represents the Precipitation of driest quarter, Bio18 represents the Precipitation of warmest quarter , Bio19 represents the Precipitation of coldest quarter.

Point 7 (L151) : I would prefer to see all of the tested bioclimatic variables including the ones that were not significant.

Response 7: Taking into account the suggestions of the reviewers, the contribution rates of all the tested bioclimatic variables has been added to the supplementary materialsof the MS. Details are as follows:

Table S1. The contribution rates of bioclimatic variables affecting the distribution of M. alternatus

Code

Bioclimatic Variables

Contribution rate/%

Bio14

Precipitation of driest month

46.4

Bio12

Annual precipitation

11.2

Bio6

Min. temperature of the coldest month

9.2

Bio10

Mean temperature of warmest quarter

7.2

Bio5

Max temperature of warmest month

5.2

Bio4

Temperature seasonality

4.8

Bio8

Mean temperature of the wettest quarter

2.8

Bio1

Annual mean temperature

2.7

Bio2

Monthly mean temperature difference

1.6

Bio19

Precipitation of coldest quarter

1.4

Bio15

Precipitation seasonality

1.3

Bio16

Precipitation of wettest quarter

1.1

Bio11

Mean temperature of coldest quarter

1

Bio7

Temperature annual range

1

Bio17

Precipitation of driest quarter

0.8

Bio9

Mean temperature of driest quarter

0.8

Bio3

Isothermality

0.8

Bio18

Precipitation of warmest quarter

0.4

Bio13

Precipitation of wettest month

0.3

Point 8 : I would put figures 3, 4, and 5 in supplementary info with more information about what the abbreviations mean – figures should be standalone

Response 8: Many thanks to the reviewers for their useful suggestions. Considering reviewer’s suggestion, Figure 3 (The ΔAICc values of the 48 different combinations of the

MaxEnt parameter settings), Figure 4 (The AUCavg values of the 48 different combinations of the MaxEnt parameter settings.), and Figure 5 (ROC curve and the values of AUC for the optimized MaxEnt model) have been moved to the supplementary materials of the MS.

Point 9 (L215) : “quarter” – repeated throughout text, please check all

Response 9: Thanks very much for reviewer’s clear advice. We have gone through the manuscript again to check the misspelling of "quarter" and corresponding modifications have been made in L18, L280, L325, and Table1.

Point 10 (L223-236) : this is just describing the figure – should be cut down

Response 10: Thanks very much for reviewer’s clear advice. Combined with your suggestions and those of another reviewer, Figure 7 (Pearson correlation analysis of insect community along the altitudinal gradient in the Guandi Mountain) has been moved to the supplementary materials of the MS. Meanwhile, the relevant explanation and description of Figure7 in L221-236 has also been deleted from MS. The details of the deleted content are as follows: Based on the response curve of bioclimatic variables, the probability of the presence of M. alternatus worldwide could be assessed using the MaxEnt model (Figure 7). The optimum value for Bio2 had a negative impact on the distribution of M. alternatus, and the response probability decreased with increasing temperature (Figure 7a). The suitable range of the minimum temperature of the coldest month (Bio6) was -25 °C to 23.15 °C, in which the response probability of M. alternatus increased with the increase in temperature from -25 °C to 10.19 °C, and decreased with the increase in temperature from 10.19 °C to 23.15 °C (Figure 7b). Mean temperature of warmest quarte (Bio10) had a positive impact on the distribution of M. alternatus, and the response probability increased with an increase in temperature from -7.46 to 36.41 °C (Figure 7c). The suitable range of the annual precipitation (Bio12) was 115.82–4404.3 mm, involving a positive relationship with the distribution of M. alternatus for precipitation below 1136.42 mm but a negative response for precipitation greater than 1136.42 mm (Figure 7d). Furthermore, the suitable range of the precipitation of driest month (Bio14) was 1.18–191.40 mm, and the response probability of M. alternatus increased for Bio14 less than 105.17 mm, whereas it decreased at greater than 105.17 mm (Figure 7e).

Point 11 (Figure 8) : Figure 8 overlay current distribution points? And have a detail showing current distribution?

Response 11: Figure 8 (now changed to Figure 3) has covered all the current 189 global distribution points of M. alternatus. Moreover, the current potential distribution area predicted by the MaxEnt model is broader than the actual distribution area of M. alternatus. For example, countries such as Vietnam, India, Pakistan, Bangladesh, Myanmar, Nepal, Bhutan, and the United States currently have no distribution areas for wild populations of M. alternatus. The above description has been added to L292-296 of the MS.

Point 12 (L335-L341) : repetition of results – rephrase / cut down and include direction?

Response 12: Thanks very much for reviewer’s clear advice. Combined with your suggestions and those of another reviewer, the relevant description in L335-341 has been deleted from MS. The details of the deleted content are as follows:

In our study, the dominant bioclimatic variables of the temperature factors affecting the potential distribution of M. alternatus were the monthly mean temperature difference (Bio2), the minimum temperature of the coldest month (Bio6), and mean temperature of warmest quarter (Bio10), and the main precipitation factors are annual precipitation (Bio12), and the precipitation of driest month (Bio14).

Point 13 (L345) : give direction of results – this is an issue throughout

Response 13: I am very sorry that I did not understand the meaning of "give direction of results" proposed by the reviewer. In order to further improve the quality of MS, it is necessary to ask the reviewer to be more specific about this problem. Thank you.

Point 14 (L394-L396) : biotic factors?

Response 14: Thanks to the helpful suggestions from the reviewers, a writing error in MS was avoided. What we want to express here are the biological factors that affect the growth and development of M. alternatus. The word “abiotic factors” has been revised to “biotic factors” in L314.

Response 15: Due to the addition and reduction of some references, the order of references in MS has been adjusted to some extent, and we have also marked them in MS accordingly.

Response 16: By reviewing the relevant literature, we know that people with red-green color blindness are able to see blue, yellow, and purple. Therefore, we have added the above three colors in Figures 6,8,9, and 10 (and now Figure 2, 3, 4, 5).

Response 17: Due to the reduction of Figures, the order of Figures in MS has been adjusted to some extent, and we have also marked them in MS accordingly.

Reviewer 3 Report

Summary

This manuscript showcases an interesting species, Monochamus alternatus, and uses MaxEnt to model the potential distribution of the species globally. The data utilized in this manuscript include new occurrence data outside of the priorly known range. The authors incorporated the 19 bioclim variables, removed highly correlated variables, and optimized the model, resulting in a high-performing model being selected by AUC. Potentially suitable areas spanned all continents, sans Antarctica, and future potential suitable areas were greater spatially over time.

General comments

This manuscript encouraged interest in the study species and prompted me to look at some more details because the system is interesting. Also, the authors do a good job of detailing their MaxEnt methods, which I appreciated as many published papers with species distribution models (SDM) do not go into as much fine detail. The modeled maps are interesting, but unfortunately, they are difficult to read due to low image resolution.

This manuscript does show that the potential distribution of M. alternatus is across several continents as the environment is suitable for the niche space modeled by the bioclim variables. However, this paper has not convinced me of two main topics. First, the model outcomes shown here is different than the others already published and cited in this manuscript. Second, the insect is capable of immigrating into other suitable environments in other countries around the world, and thus, understanding its worldwide potential distribution is important. Both topics can easily be introduced in the introduction and fully addressed in the discussion.

Specific comments

Line 40: It is a formal procedure to provide the species order, family, and author information for your species. A common name may also help the reader better understand what insect you are referring to.

Figure 1: This map appears to be in low resolution and is difficult to look at. As you have stated, the insect is only in the eastern region of Asia, so a zoomed-in visual of the area would help clarify the exact distribution. Furthermore, this figure could easily provide additional visual information. For example, you could showcase which countries have classified the insect (mentioned in line 51) or which location points were gathered in which year (lines 62-66).

Lines 95-105: This paragraph could use a little clarification. Its purpose appears to be a reference to previously published maps of M. alternatus potential distribution and how yours differs. However, I had to read the paragraph a few times to understand that your maps will provide new knowledge beyond your 4 references. I believe the new location points that you mentioned in lines 62-66 are what is novel. If this is the case, adding this to your Figure 1 caption in addition to the map suggestions above would help illuminate the importance of your work.

Line 145: How did you choose which correlated variable to keep?

Figure 2: Pearson test results are generally not displayed in SDM MaxEnt papers. This could easily be moved to an appendix to provide more space for larger and higher-resolution maps. A key or text in the figure caption naming each variable would be helpful. Also, the light-colored font is harder to read. Perhaps color coding with a threshold instead of continuously would help limit this problem.

Figure 4: Another figure that could move to an appendix to make room. Also, the line colors are too similar to differentiate.

Figures 6,8,9,10: You should change the color scheme to allow those with red-green blindness to read the figures.

Figures 8,9,10: These maps would benefit from a much higher resolution. They are difficult to understand.

Line 269: What is the difference between these two climate scenarios? This was not discussed in the methods, nor is there an explanation in the results. It would be easier to understand the changes in your maps if this is explained somewhere.

Line 305: Please clarify what you mean by “lates”.

Lines 307-312: This was stated in lines 78-88 and does not seem necessary to repeat here.

Lines 327-329: Did you run the data through MaxEnt without the additional samples as well? If not, then this statement is not necessarily true as the increase in AUC values may also be due to the species distribution model (SDM) utilized.

Lines 334-337: The only variables that were included in the model were different values of temperature and precipitation as those are the bioclimatic variables. With this in mind, the statement “The results of this study confirm these findings” is expected. Consider rephrasing this section to be more informative.

Line 340 and 348: Misspelling – quarte to quarter

Lines 355-357: This sentence was just stated in lines 340-341. As the statements concerning precipitation are in these later lines of the paragraph, removing the statement from 340-341 would help reduce redundancy.

Lines 361-372: A majority of this section is a repeat of the results without providing any conclusions or commentary. This is the case throughout the discussion section. Selectively removing results from the discussion would streamline the reading, reduce redundancy, and help showcase any conclusions much more effectively.

Lines 394-395: The examples provided are biotic, not abiotic factors. 

Author Response

Thanks so much for reviewer 3 for his/her positive comments on our work. We have revised the manuscript as reviewer mentioned. Those comments are all valuable and very helpful for revising and improving our manuscript, as well as the important guiding significance to our researches. We have studied the comments carefully and have revised and corrected the manuscript which we hope meet his/her approval. Revised portions are marked in yellow in the revised manuscript. The responses are as following:

Point 1 (L40) : It is a formal procedure to provide the species order, family, and author information for your species. A common name may also help the reader better understand what insect you are referring to.

Response 1: Thanks very much for reviewer’s useful advices. Considering reviewer’s suggestion, the common name, author information, and taxonomic information (Pine sawyer beetle, Monochamus alternatus Hope (Coleoptera: Cerambycidae)) of the M. alternatus have been added to the original sentence in L39.

Point 2 (Figure 1): Figure 1 appears to be in low resolution and is difficult to look at. As you have stated, the insect is only in the eastern region of Asia, so a zoomed-in visual of the area would help clarify the exact distribution. Furthermore, this figure could easily provide additional visual information. For example, you could showcase which countries have classified the insect (mentioned in line 51) or which location points were gathered in which year (lines 62-66).

Response 2: Thanks very much for reviewer’s useful advices, the resolution of Figure 1 has been improved in the MS. We tried to make a zoomed-in Figure of the global distribution of M. alternatus, but the effect of the picture is not very ideal. Therefore, we decided to keep the Geographical distribution sampling point of M. alternatus at global scale in MS. In this way, we have a better understanding of the global distribution of M. alternatus.

Point 3 (L95-L105) : This paragraph could use a little clarification. Its purpose appears to be a reference to previously published maps of M. alternatus potential distribution and how yours differs. However, I had to read the paragraph a few times to understand that your maps will provide new knowledge beyond your 4 references. I believe the new location points that you mentioned in lines 62-66 are what is novel. If this is the case, adding this to your Figure 1 caption in addition to the map suggestions above would help illuminate the importance of your work.

Response 3: Thanks very much for reviewer’s useful advices. Indeed, as the reviewer said, what we want to express in this paragraph is that the difference between this study and the previous published studies of M. alternatus potential distribution. In Figure 1, the new occurrence geographical distribution sampling point of M. alternatus has been marked with blue color, and the corresponded explanation has been added in the caption of Figure 1.

Point 4 (L145) : How did you choose which correlated variable to keep?

Response 4: In this study, 19 historical global bioclimatic variables that are closely related to the growth and development of M. alternatus were downloaded from the worldclimate database. In order to avoid the autocorrelation of multiple linear repeats between the extracted climate variables and the overfitting of MaxEnt, we need to screen and eliminate part of the climate variables to reduce the influence of redundant information on the simulation results. First, the extraction analysis tool in ArcGIS software was used to perform the multicollinearity analysis of each climatic variable; then, the data were analyzed using Pearson correlation in SPSS 22.0. Climatic variables with correlation values greater than 0.8 were removed to improve the accuracy of the model simulation and minimize the impact of over-fitting.

Point 5 (Figure 2) : As for Figure 2, Pearson test results are generally not displayed in SDM MaxEnt papers. This could easily be moved to an appendix to provide more space for larger and higher-resolution maps. A key or text in the figure caption naming each variable would be helpful. Also, the light-colored font is harder to read. Perhaps color coding with a threshold instead of continuously would help limit this problem.

Response 5: Many thanks to the reviewers for their useful suggestions. Considering reviewer’s suggestion, Figure 2 (Pearson correlation analysis and correlation coefficients of 19 bioclimatic variables) has been moved to the supplementary materials of the MS.

Point 6 (Figure 4) : For Figure 4, Another figure that could move to an appendix to make room. Also, the line colors are too similar to differentiate.

Response 6: Many thanks to the reviewers for their useful suggestions. Considering reviewer’s suggestion, Figure 4 (The AUCavg values of the 48 different combinations of the MaxEnt parameter settings) has been moved to the supplementary materials of the MS.

Point 7 (Figure 6,8,9,10) : For Figures 6,8,9,10, You should change the color scheme to allow those with red-green blindness to read the figures.

Response 7: Thanks very much for reviewer’s useful advices, it was our mistake that not to take it into account. By reviewing the relevant literature, we know that people with red-green color blindness are able to see blue, yellow, and purple. Therefore, we have added the above three colors in Figures 6,8,9, and 10 (and now Figure 2, 3, 4, 5).

Point 8 (Figure 8,9,10): For Figures 8,9,10, These maps would benefit from a much higher resolution. They are difficult to understand.

Response 8: Thanks very much for reviewer’s useful advices, the resolution of Figure 8,9,10 (and now Figure 3, 4, 5) have been improved in the MS.

Point 9 (L269) : What is the difference between these two climate scenarios? This was not discussed in the methods, nor is there an explanation in the results. It would be easier to understand the changes in your maps if this is explained somewhere.

Response 9: Thanks very much for reviewer’s useful advices. We chose two scenarios available in the WorldClim 2.1 database. These scenarios reflect uncertainties in possible trajectories of climate change mitigation with SSP126 belonging to the low-forcing scenario and SSP585 belonging to the high-forcing scenario. Relevant explanations for SSP126 (sustainability, the most optimistic scenario reflecting RCP2.6 from the fifth report) and SSP585 (fossil-fuel based development or business-as-usual, reflecting RCP8.5) have been added to L32-134.

Point 10 (L305) : Please clarify what you mean by “lates”.

Response 10: Thanks to the helpful suggestions from the reviewers, a writing error in MS was avoided. The word “lates” has been revised to “latest” in L360.

Point 11 (L307-L312) : This was stated in lines 78-88 and does not seem necessary to repeat here.

Response 11: Considering reviewer’s suggestion, we decided to remove this part from MS. The details of the deleted content are as follows:

Based on the maximum entropy theory, the MaxEnt model simulates and analyzes the distribution of species when the entropy reaches a maximum under restricted conditions, according to the distribution of the known species and the corresponding environmental variables. This model has the advantages of high simulation accuracy, short operation time, and easy operation and has been widely used in the prediction of potentially suitable areas for plants, insects, and other organisms.

Point 12 (L327-L329) : Did you run the data through MaxEnt without the additional samples as well? If not, then this statement is not necessarily true as the increase in AUC values may also be due to the species distribution model (SDM) utilized.

Response 12: Thanks very much for reviewer’s useful advices. The statement of “The prediction accuracy of the MaxEnt model was measured using the AUC value, and the prediction accuracy of the model increased with an increase in the number of species samples” in L327-329 is not accurate enough because we did not use additional samples to analyze the accuracy of the MaxEnt model. Therefore, we have removed this sentence from MS to avoid ambiguity.

Point 13 (L334-L337) : The only variables that were included in the model were different values of temperature and precipitation as those are the bioclimatic variables. With this in mind, the statement “The results of this study confirm these findings” is expected. Consider rephrasing this section to be more informative.

Response 13: Considering reviewer’s suggestion, the sentence “The results of this study confirm these finding” has been removed from the MS.

Point 14 (L340andL348) : Misspelling – quarte to quarter

Response 14: We are sorry that this is a mistake in the spelling of the word, thanks very much for reviewer’s useful advices. We have gone through the manuscript again to check the misspelling of "quarter" and corresponding modifications have been made in L18, L280, L325, and Table1.

Point 15 (L355-L357) : This sentence was just stated in lines 340-341. As the statements concerning precipitation are in these later lines of the paragraph, removing the statement from 340-341 would help reduce redundancy.

Response 15: Considering reviewer’s suggestion, we decided to remove “In the present study, the main precipitation variables affecting the distribution of M. alternatus were the annual precipitation (Bio12) and the precipitation of driest month (Bio14)” from MS.

Point 16 (L361-L372) : A majority of this section is a repeat of the results without providing any conclusions or commentary. This is the case throughout the discussion section. Selectively removing results from the discussion would streamline the reading, reduce redundancy, and help showcase any conclusions much more effectively.

Response 16: Thanks very much for reviewer’s clear advice. Combined with your suggestions, we removed sentences in L361-369 from MS in order to reduce redundancy. The details of the deleted content are as follows:

Based on 189 distribution record points and 8 bioclimatic variables, the current and future potential distribution prediction maps of M. alternatus were established according to four grade standards of high suitability, moderate suitability, low suitability, and poor suitability areas. Under the current climate conditions, the potentially suitable habitat areas of M. alternatus were distributed across all continents except Antarctica, accounting for 4.17% of the total land area of the world. The total potential suitable habitat areas can be divided as follows: potentially highly suitable areas accounting for 23.72%, potentially moderately suitable areas accounting for 15.13%, and potentially low-suitability areas accounting for 61.15%.

Point 17 (L394-L395) : The examples provided are biotic, not abiotic factors.

Response 17: Thanks to the helpful suggestions from the reviewers, a writing error in MS was avoided. What we want to express here are the biological factors that affect the growth and development of M. alternatus. The word “abiotic factors” has been revised to “biotic factors” in L314.

Response 18: Due to the addition and reduction of some references, the order of references in MS has been adjusted to some extent, and we have also marked them in MS accordingly.

Response 19: Due to the reduction of Figures, the order of Figures in MS has been adjusted to some extent, and we have also marked them in MS accordingly.

Round 2

Reviewer 1 Report

"Response 6:" The "TSS" stands for 'True Skill Statistics ' please see "

Assessing the accuracy of species distribution models: prevalence, kappa and the true skill statistic (TSS)"  by 

Allouche et. al .         https://besjournals.onlinelibrary.wiley.com/doi/10.1111/j.1365-2664.2006.01214.x